# Study on *ZmRPN10* Regulating Leaf Angle in Maize by RNA-Seq

**DOI:** 10.3390/ijms24010189

**Published:** 2022-12-22

**Authors:** Xiangzhuo Ji, Bingbing Jin, Zelong Zhuang, Fangguo Chang, Fang Wang, Yunling Peng

**Affiliations:** 1College of Agronomy, Gansu Agricultural University, Lanzhou 730070, China; 2Gansu Provincial Key Laboratory of Aridland Crop Science, Gansu Agricultural University, Lanzhou 730070, China; 3Gansu Key Laboratory of Crop Improvement & Germplasm Enhancement, Lanzhou 730070, China

**Keywords:** maize (*Zea mays* L.), *ZmRPN10*, leaf angle, RNA-Seq

## Abstract

Ubiquitin/proteasome-mediated proteolysis (UPP) plays a crucial role in almost all aspects of plant growth and development, proteasome subunit RPN10 mediates ubiquitination substrate recognition in the UPP process. The recognition pathway of ubiquitinated UPP substrate is different in different species, which indicates that the mechanism and function of *RPN10* are different in different species. However, the homologous *ZmRPN10* in maize has not been studied. In this study, the changing of leaf angle and gene expression in leaves in maize wild-type B73 and mutant *rpn10* under exogenous brassinosteroids (BRs) were investigated. The regulation effect of BR on the leaf angle of *rpn10* was significantly stronger than that of B73. Transcriptome analysis showed that among the differentially expressed genes, CRE1, A-ARR and SnRK2 were significantly up-regulated, and PP2C, BRI1 AUX/IAA, JAZ and MYC2 were significantly down-regulated. This study revealed the regulation mechanism of *ZmRPN10* on maize leaf angle and provided a promising gene resource for maize breeding.

## 1. Introduction

Maize is the most important food crop and industrial raw material in the world. Leaf angle is an important agronomic trait determining maize planting density and light penetration into the canopy and contributes to the yield gain in modern maize hybrids [1]. The size of the leaf angle in plants is one of the important parameters affecting photosynthetic efficiency and canopy interception [2]. The size of the leaf angle depends on the cell division, expansion, and cell wall composition of the pillow part (including ear and tongue) connecting the leaf and sheath tissue. Plants with smaller leaf angles have better canopy structures, can intercept more light energy, maximize photosynthetic efficiency, and significantly impact seed yield [3,4].

Recent studies have found that the development of the leaf angle is jointly regulated by various plant hormones [5,6,7]. Other studies have concluded that mutant plants lacking or insensitive to BR have more upright leaves. Overexpression of BR biosynthetic genes or signal transduction will lead to fewer leaves and a reduced leaf angle [8,9]. Rice lines overexpressing *OsARF19* showed increased leaf inclination due to increased paraxial cell division. Analysis of expression mode showed that the *OsARF19* gene was expressed in various organs, such as the rice leaf angle, and was strongly induced by BR [10]. The maize genes INFLORESCESCENCE4 (*BIF1* and *BIF4*) regulate the early steps required for inflorescence formation. *BIF1* and *BIF4* encode the auxin / indole-3-acetate ACID (Aux / IAA) proteins, which are key components of the auxin hormone signaling pathway and are essential for organogenesis [11]. The candidate gene *ZmACS7* was identified in maize, and alteration of the C-terminal phase of *ZmACS7* in the mutant resulted in increased stability of this encoded protein. The *ZmACS7* plays a key role in the number of leaves, root development, and regulation of flowering time in maize [12]. According to the gene *OsTAC1* regulating leaf angle in rice, we successfully cloned the gene *ZmTAC1*, which is highly homologous in maize, and preliminarily verified the relationship between *ZmTAC1* and leaf angle in maize [13,14]. According to previous findings, *ZmCLA4* directly binds to the promoters of many genes and regulates leaf angle development in plants by forming a complex regulatory network with various hormones [15].

The 26S proteasome plays vital role in regulating plant growth and development, participating in various signal transduction processes and cell cycle regulation [16]. *RPN10* is a subunit of the 26S proteasome, which can recognize polyubiquitinated proteins and widely exists in the cytoplasm and nucleus [17], and two ubiquitin receptors that form proteasome replacement pathways in yeast include *RPN10* and *RAD23* [18]. Researchers have found that the change of the *RPN10* subunit makes the *Arabidopsis* mutant plants show the characteristics of seed germination, growth rate, stamen number, male gamete genetic transmission and hormone-induced cell division reduction, and reduced sensitivity to cytokinin and auxin [19]. In Arabidopsis, *RPN12* participates in response to cytokinin, and the T-DNA insertion of *RPN12a* into the mutant reduces the leaf formation rate, root elongation, and stem morphology [20]. The rice homologues of the yeast *RP* subunit gene were identified from the rice EST library, their nucleotide sequences were determined, and their products were found to be assembled in the rice *RP* complex [21].

A mutant *T429* with a large leaf angle was isolated from the mutant library of transgenic lines with T-DNA inserted into rice, and the gene *OsWRKY11* regulating leaf angle was successfully cloned [22]. Studies have shown that the ovalbumin family protein 6 (OsOFP6) in rice is a positive regulator, regulating the expression of genes related to the biosynthesis of the secondary cell wall. The transgenic plants that knock out the *OsOFP6* gene through RNA interference, due to the thinning of the secondary cell wall and the reduction in the content of cellulose and lignin, lead to the increase of leaf angle [23]. In this study, we identified a *ZmRPN10* mutant *rpn10* from an ethyl methane sulfonate (EMS) mutant library of maize inbred line B73, which displays a relatively larger leaf angle. With B73 and *rpn10* as experimental materials, we discussed the regulation mechanism of *ZmRPN10* on maize leaf angle from field characteristics, cytological analysis, bioinformatics, expression pattern analysis, exogenous BR response and transcriptome analysis. It is expected to provide help for breeding ideal plant types of leaf angle and planting innovations of excellent plant types.

## 2. Results

### 2.1. Identification of ZmRPN10 Gene

The complete sequence of the *ZmRPN10* (GRMZM2G147671) gene of wild type B73 was obtained from the NCBI database, which is located on chromosome 9, full-length 4352bp, including 10 exons and 9 introns, encoding 401 amino acids. The mutation of *rpn10* occurs at the beginning of the third exon of the *ZmRPN10* gene sequence, and the base changes from G to A. The mutation site was splicing the *ZmRPN10* gene of the receptor, which led to the increase of the B73 leaf angle. (Figure 1A). The protein sequence of ZmRPN10 was compared and analyzed by Blastp in NCBI (GeneBank). It was found that in the plant kingdom, ZmRPN10 had 98.5% homology with *Sorghum bicolor* (XP_002465571.1), 94.81% homology with *Panicum_miliaceum* (RLN40811.1), 94.07% homology with *Panicum_miliaceum* (XP_004985037.1) and 95.52% homology with *Panicum_miliaceum* (XP_004985038.1), and 84.16% homology with *Hordeum_vulgare* (KAE8779376.1). It has 84.87% homology with *Triticum aestivum* (KAF7049770.1), 63.28% homology with *Arabidopsis thaliala* (XP_020872407.1), 63.95% homology with *Arabidopsis thaliala* (NP_195575.1), 87.84% homology with *Oryza sativa* (XP_015632315.1), 72.66 homology with *Cucurbita maxima* (XP_022965261.1), and 74.74% homology with *Dioscorea cayenensis* (XP_039138573.1). The protein homology with *Cucumis sativus* (XP_004146147.1), *Ziziphus jujuba* (XP_015889728.1) and *Sesamum indicum* (XP_011099843.1) was 70.37%, 72.48%, and 73.02%, respectively. Use mega software to compare the amino acid sequences of the above species and build an evolution tree. It is found that ZmRPN10 is the closest relative to *Sorghum bicolor* (XP_002465571.1) (Figure 1B). RNA was extracted from the roots, stems, the first pulvinus, first leaf, the second pulvinus and second leaf tissues of B73 seedlings in the first week, second week, and third-week growth stages after planting. qRT-PCR was performed after reverse transcription. The results are as follows: under three different periods, the expression level in the roots was the lowest, and with the increase of seedlings, the expression level in the pulvinus gradually increased. After three weeks, the relative expression of the first leaf occipital part was the highest, and there was no significant change in other parts (Figure 1C).

### 2.2. Phenotypic Analysis of B73 and rpn10

Field phenotype comparisons of the B73 and *rpn10* materials revealed significant differences across multiple traits (Figure 2 and Table 1). Compared with B73, the plant height of *rpn10* was significantly reduced by 10.0%, and the spike height of *rpn10* was 13.8% lower than that of B73. According to the observation of ear transection cytology, the thickness of *rpn 10* leaf ear is larger than B73, with an average thickness increase of 74.2%. The parenchyma cell size of *rpn 10* leaf ear is larger than B73, and the average size increased by 111.5%. The comparative analysis of maize grains harvested at maturity showed that the grain characteristics changed significantly. At the powder dispersing stage, except for the fifth leaf, the angle between leaves of *rpn10* was greater than that of B73, and the average angle of three Ear-leaves increased by 16.1%. The main spike length of *rpn10* tassels was 16.2% lower than that of B73, and the number of tassel branches was 15.7% lower. The leaf length of *rpn10* increased by 18.5%, the leaf area increased by 25.4%, the leaf orientation value decreased by 10.7%, and the 100-grain weight decreased by 16.6%.

### 2.3. Analysis of the Response of rpn10 to Exogenous BR

BR plays an important role in the growth and development of corn leaf angle and the degree of leaf inclination. Applying different concentrations of BR treatment *rpn10* and B73, the leaf angle changes significantly, with the increase in BR concentration, the leaf angle first increased and then decreased. Without the application of exogenous BR, the *rpn10* leaf angle is greater than B73, reaching the maximum value at a concentration of 0.2μmol/L, the *rpn10* leaf angle is 54.26°, while B73 leaf angle is 41.3°, *rpn10* and B73 increased by 26.7% and 42.6% at 0.2μmol/L compared with 0μmol/L, respectively, indicating that *rpn10* is more sensitive to external BR (Figure 3A,B). Under 0.2 μmol/L BR treatment, the relative expression of the *ZmRPN10* in the occipital part of B73 leaves of maize increased and increased by 54.4% under exogenous BR treatment, indicating that *ZmRPN10* played a positive regulatory role in BR (Figure 3C). After the application of exogenous BR, the seedling growth of B73 and *rpn10* both increased first and then decreased, and the B73 seedling growth was more sensitive to exogenous BR response, which increased by 24.2% compared with the control group at the optimal concentration, and the maximum seedling length of *rpn10* reached the maximum at a concentration of 0.2μmol/L, which was 16.8% higher than that of the normal group. The root growth trend was consistent with that of seedlings, and there was no significant difference between the two materials at the same concentration (Figure 3D,E).

### 2.4. Venn Diagram Analysis of DEGs in B73 and rpn10 under BR Treatment

In order to investigate the change in leaf angle of B73 and *rpn10* under exogenous BR treatment of 0.2 μmol/L, we determined the fold change in DEGs (|expression under these conditions| >1 and *p*-value < 0.05). Using FPKM as a measure of gene expression, a total of 17778 genes were identified in wild-type B73 and mutant *rpn10* under control and 0.2 μmol/L exogenous BR (Figure 4). The Upset diagram reflects the distribution of DEGs in B73 and *rpn10* under BR treatment compared with CK treatment. Under 0.2 μmol/L BR treatment, 13078 DEG were identified in B73 (7483 up-regulated and 5595 down-regulated) (Appendix A). 8309 DEGs identified in *rpn10* (4528 up-regulated and 3781 down-regulated) (Appendix A). A total of 2571 DEGs were identified by B73 and *rpn10* under BR treatment, of which 207 were up-regulated, and 279 were down-regulated (Figure 4B,C). These shared DEGs may play an important role in the leaf angle tuning of maize responses to exogenous BR, reflecting differences in and B73 and in response to exogenous BR.

### 2.5. Gene Ontology Classification and Pathway Enrichment Analysis of DEGs in the Leaf Angle between B73 and rpn10 under Different

In order to further understand the biological functions of the genes in these modules, we performed both the GO and KEGG enrichment analyses (Figure 5). GO terms include categories of biological processes, cellular components, and molecular functions of a given gene product. In material B73-CK vs. *rpn10*-CK, aerobic oxidation-reduction process and phosphorylation, the cell components were membranes and integral components of membrane, and the molecular functions were metal ion binding and DNA binding (Figure 5A). The main biological processes enriched in materials B73-CK vs. B73-BR were aerobic oxidation-reduction process and phosphorylation, the cell components were membrane and integral component of membrane, and the molecular functions were ATP binding and transferase activity (Figure 5B). The main enriched biological processes, mainly between *rpn10*-CK vs. *rpn10*-BR, were aerobic phosphorylation, regulation of transcription, DNA templated, the cell components were membrane and integral component of membrane, and the molecular functions were transferase activity and ATP binding (Figure 5C). The results showed that the Gene-Ontology Classification, genes regulating the leaf angle between B73 and *rpn10*, varied greatly under exogenous BR treatment. The number of genes annotated by metabolic pathways under different comparison groups and the number of enriched pathways are also not the same. For bubble plots comparing group top20 enrichment pathways. Between Material B73-CK vs. *rpn10*-CK, the most enriched pathways were Plant hormone signal transduction, Photosynthesis-antenna proteins and more genes were enriched in Phenylpropanoid biosynthesis, the MAPK signaling pathway-plant and Protein processing in the endoplasmic reticulum (Figure 5D). The more enriched pathways in the B73-CK vs. B73-BR comparison group were Glycerolipid metabolism and Glycolysis/Gluconeogenesis (Figure 5E). Pathways enriched for a higher number of genes in the *rpn10*-CK vs. *rpn10*-BR comparison group were also Glutathione metabolism and Sulfur metabolism (Figure 5F).

### 2.6. Gene Co-Expression Network Analysis

WGCNA (Weighted correlation network analysis) is a systems biology approach used to describe gene association patterns between different samples. It can be used to identify highly synergistic gene sets and identify alternate biomarker genes based on the cohesion of gene sets and the association between gene sets and phenotypes. We performed transcriptome sequencing of 12 samples (B73 and *rpn10*, 2 treatments and 3 biological replicates), filtered out genes with mean FPKM expression below 1, had a total of 19,076 genes, and set the similarity threshold of Fold > 0.5 for Module fusion to build a co-expression network module (Figure 6A,B). Based on the correlation between B73 and *rpn10* phenotype traits with exogenous BR treatment, 14 modules were associated with leaf angle phenotype, and we screened for modules that were significantly correlated with phenotype, and we observed that maize leaf angle change was significantly positively correlated with the grey60 module (0.7), the ivory module (0.69), and the light green module (0.56) (Figure 6C,D).

### 2.7. Functional Analysis of Genes in the Leaf Angle Core Module

We performed the GO and KEGG enrichment analysis on the genes within the grey60 module and light green module that was significantly associated with the increase of the maize leaf angle and analyzed their main biological functions (Figure 7). The GO biological process phosphorylation is enriched in the Grey60 module, and the cellular components are membrane and plasma membrane, and the molecular functions are kinase activity and ATP binding (Figure 7A). KEGG enrichment analysis indicated that the genes in the grey60 module are most frequently involved in starch and sucrose metabolism, MAPK signaling pathway in plants (Figure 7B). The enriched GO biological processes regulation of transcription, DNA templated, phosphorylation, and cellular component are membrane and integral components of membrane, and the molecular functions are protein binding and kinase activity (Figure 7C). KEGG enrichment analysis showed that the genes in the light green module are most frequently involved in Plant hormone signal transduction, and protein processing in the endoplasmic reticulum (Figure 7D). Therefore, we speculate that these pathways may play an important role in regulating the angle size process of maize leaves.

### 2.8. Analysis of Network Regulation of Leaf Angle Core Module

In order to identify the specific genes most likely to be most important in regulating the maize leaf angle, we also used expression data to generate co-expression networks. The Gene construction interaction network map was filtered based on the weight range of the two grey60 and light green modules. The five genes each with the highest kME value (feature gene connection) in the module were classified as central genes. The central gene of grey60 module Zm00001d045395 hexose carrier protein HEX6, Zm00001d003174 Short-chain dehydrogenases/reductases (SDR), Zm00001d002017 Protein of unknown function (DUF538), Zm00001d042143 glucan endo-1, The 3-beta-glucosidase homolog 1 and Zm00001d045397 are U-box domain-containing protein 35 (Figure 8A). The light green module central gene is Zm00001d051799 for nudix hydrolase 4, Zm00001d009354 and 3-ketoacyl-CoA synthase, Zm00001d004451 for glyoxylate/hydroxypyruvate reductase HPR3, Zm00001d023941 for Major Facilitator Superfamily (MFS), Zm00001d048344 are Rare lipoprotein A (RlpA) -like double-psi beta-barrel (DPBB 1) (Figure 8B). In the grey60 and lightgreen module, we found both the auxin-responsive gene SAUR71 (ENSRNA049461765) and the cytokinin-responsive regulator ARR9 (Zm00001d032919). Therefore, we speculate that protein phosphorylation, an integral component of the membrane, and kinase activity are important in the enlargement of maize leaf angle, and that auxin and cytokinin are also essential.

### 2.9. Validation of DEGs by qRT-PCR Analysis

Seven differentially expressed genes were selected at B73 and *rpn10* for qRT-PCR validation, and the results showed that (Figure 9). Zm00001d044183 and Zm00001d037547 genes were higher in *rpn10* than B73, Zm00001d018414 was lower than B73, Zm00001d039520 difference was not obvious, Zm00001d020705, Zm00001d020332 and Zm00001d004506 were reduced in *rpn10* and increased in B73, these results are consistent with transcriptome sequencing results, indicating reliable sequencing results.

## 3. Discussion

### 3.1. ZmRPN10 Mutation Affects the Phenotype of Maize Plants

The genome-wide mutation collection of gene indexes obtained by using B73 pollen with EMS treatment provides an essential resource for the functional analysis of maize genes and brings ideal allelic variation for genetic breeding in maize [24]. Using mutant *vks1* studies with different grain size phenotypes, *vks1* encoding *ZmKIN11* belongs to the kinesin-14 subfamily and plays key roles in the migration of free nuclei in the coenocyte as well as in mitosis and cytokinesis in early mitotic divisions [25]. As a study of *bm1*, a mutant of *ZmCAD2* of the cassia bark alcohol dehydrogenase gene, Mutations in *ZmCAD2* significantly reduced its transcription rate and enzymatic activity, resulting in reduced lignin content and changes in lignin composition [26]. The mutant *ila1* with increased leaf angle was obtained by T-DNA. The increase in the leaf angle of *ila1* was caused by the decrease in the mechanical strength of the leaf occipital part [27]. In this study, we found significant differences between *rpn10* and B73, with *rpn10* leaf angle, leaf angle increasing by 16.1%, and leaf width by 18.5%. Compared with B73, *rpn10* decreased by 10.0%, 16.2%, and 15.7% in plant height, male ears with branches, and male ears with branches over B73, respectively. On grain traits, the mutant showed a significant reduction in spike length and single spike weight, and the grain was oblate.

### 3.2. Plant Gene Mutations Affect Leaf Angle Changes

Leaf angle is a key trait of plant architecture and a target for the genetic improvement of crops [5]. A study found that transcription factor (TF) *ZmBEH1* (BZR1/BES1 homolog gene 1) is targeted by *ZmLG2* and regulates leaf angle formation by influencing sclerenchyma cell layers on the adaxial side [28]. Maize *YABBY* mutant drooping leaf1 (*drl1*) affects the leaf length and width, leaf angle, internode length and diameter, leading to the sagging and increased angle of the *drl1* mutant leaves [29]. For wheat *TaSPL8* mutant compact plant architecture (*cpa*) flag leaf angle junction by electron microscope scanning results shows that the mutant has a larger leaf angle that is mainly increased by the number of cells, with cell size showing no obvious difference [30]. The transcriptomic data revealed that the expression of *ZmBEH1* and *ZmLG2* were down-regulated at the pulvinus of B73 under 0.2μmol/L BR treatment conditions, and the expression of *YABBY* did not change. The results indicated that the change of B73 pulvinus was correlated with *ZmBEH1* and *ZmLG2* under BR treatment. In this experiment, we observed the cytology of B73 and *rpn10* ears at the powder dispersing stage by paraffin section. It was found that the thickness of *rpn10* ear was significantly greater than B73, the number of layers of the *rpn10* cells was more than B73, the cell volume of *rpn10* was greater than B73, and the *rpn10* cells were more scattered and more than B73. Therefore, we speculated that the increased leaf angle of *rpn10* might be caused by the increased cell volume and number of cells at the pulvinus.

### 3.3. Analysis of Genes Related to the Regulation of Maize Leaf Angle under Exogenous BR Treatment

Transcriptome sequencing can quickly and comprehensively obtain gene expression changes in different tissues or organs under specific conditions, and differential gene analysis is helpful to analyze gene regulatory networks [31]. Transcriptome sequencing of B73 and *lpa1* plays an important role in mining functional genes related to different traits [32]. To investigate the effect of exogenous BR on maize leaf angle. In this experiment, the materials B73 and *rpn10* were used as materials, based on the results of the transcriptome sequencing, with |log2FC| > 2 and *p* < 0.001, screened out of 18 DEGs (Appendix A), These genes are mainly involved in the plant protein ubiquitination, gibberellin biosynthetic process, protein autophosphorylation and ATPase activity et al. Among which Zm00001d033100 encodes the *ATPS1* gene, study show that the *ATPS1* expression quantity difference affects the total ATP sulfurylase (*ATPS*) activity in plants and results in a limited capacity to reduce sulfate and variation in sulfate level [33]. Zm00001d003068 encodes the *PPCK2*, *ZmPPCK2* is expressed in leaf bundle sheath cells, preferentially in the dark, the *ZmPPCK2* product is to allow PEPC to function anaplerotically in bundle sheath cells in the dark without interfering with the C4 cycle [34]. Zm00001d020332 encodes the Ubiquitin-protein ligase, the E3 ubiquitin ligase in UPP is responsible for specific recognition of target proteins and ubiquitination modifications that enable it to maintain multiple biological functions in cell division, signal transduction, adversity stress, growth and development [35,36]. Zm00001d012212 encodes the gibberellin 20-oxidase5 (*GA20ox5*), *GA20ox* have been shown before to be implicated in a number of GA-controlled processes such as stem elongation, flower formation, and fruit growth [37]. In Arabidopsis, *AtGA20ox1* and *AtGA20ox2*, control redundantly various processes such as the hypocotyl and internode elongation, with *AtGA20ox1* controlling mostly the internode and filament elongation and stem elongation [38]. Zm00001d011988 encodes the vacuolar protein sorting-associated protein 41 homolog, in plants the secretory and biosynthetic trafficking pathways are involved in a series of vital mechanisms, such as gravitropism, autophagy, hormone transport, cytokinesis and abiotic/biotic stress responses [39]. Zm00001d048520 encodes the Aquaporin tonoplast intrinsic aquaporin 3 (*TIP3*). Aquaporins have important roles in various physiological processes in plants, including growth, development and adaptation to stress, TIP coding genes have been shown to be responsive to salt or drought stress in different plant species [40]. Therefore, we can speculate that under exogenous BR treatment, *ZmRPN10* may regulate maize leaf angle changes through these differentially expressed genes.

### 3.4. Effect of ZmRPN10 on Plant Hormone Signal Transduction Pathway

The development of leaf angle is regulated by many factors, and plant hormones play an important role in regulating the development of leaf angle. By analyzing the DEGs of B73 and *rpn10*, we found that three genes regulating CYTOKININ RECEPTOR 1 (CRE1), A-ARR and SNF1-RELATED KINASE 2 (SnRK2) were significantly up-regulated, and five genes regulating PP2C, BRASSINOSTEROID INSENSITIVE1 (BRI1), AUX/IAA, JASMONATE ZIM-DOMAIN (JAZ) and MYELOCYTOMATOSIS PROTEIN 2 (MYC2) were significantly down-regulated in plant hormone signal transduction pathway. In plants, CRE1 acts as a regulatory gene for CTK signaling binds to extracellular CTK, and phosphorylates itself, which is a key step in signal transduction [41]. A-ARRs are negative regulators of cytokinin signal transduction. They can play a negative feedback role in regulating cytokinin signal transduction by competing for the phosphate transporter proteins (AHPs) of B-ARRs and inhibiting the transcription of B-ARRs [42]. SnRK2 belongs to a subfamily of SnRK and is a kind of plant-specific serine/threonine protein kinase. This gene family is highly conservative in evolution and widely exists in plants. It is the hub of plant growth and development and the stress response regulation network [43,44]. PP2C is a kind of monomer serine/threonine residue protein phosphatase. In eukaryotes, PP2C plays an important role in regulating abscisic acid, jasmonic acid, salicylic acid and other signal transduction pathways [45]. BRI1 is a receptor-like kinase with a special structure. It contains leucine repeats and is found to be the receptor protein of BR. It is crucial in the BR signal pathway. After BRI1 is activated, it will cause phosphorylation of intracellular proteases, thus causing a series of cascade reactions. Finally, it will activate the functions of some transcription factors, change the expression of some genes related to growth and development, and then change plants’ growth and living conditions [46,47]. Aux/IAA and auxin response factor (ARF) form a dimer, thereby inhibiting the transcriptional regulatory function of ARF exercise. ARF-type transcription factors can bind to the cis-acting elements of auxin-induced genes to regulate their transcriptional expression [48]. JAZ protein is one of the most important components of the JA signal pathway. As a JA common receptor, it and COI1 respond to the JA signal by inhibiting the activity of downstream transcription factors [49]. MYC2 plays an important role in the JA signal transduction pathway in plants. It is found in Arabidopsis that the MYC2 transcription factor plays a regulatory role by forming COI1/JAZs/MYC2 complex and participates in JA, ABA, and other hormone signal transduction processes [50,51]. Therefore, we developed a model in which *ZmRPN10* regulates maize leaf angle through the regulation of phytohormone signaling (Figure 10).

## 4. Materials and Methods

### 4.1. Identification and Phenotype Analysis of rpn10

Wild-type B73 was provided by the Corn Research Group, College of Agronomy, Gansu Agricultural University. Mutant *rpn10* (EMS3-091e50) was obtained from the maize EMS mutant library of Qilu Normal University (http://elabcaas.cn/memd/ (accessed on 27 July 2022)). Mutant *rpn10* was planted in the experimental field, after consecutive backcrossing of B73 for two generations using the mutant *rpn10*, then the selfing generation cleared the mutant background, and each generation using Sanger examined the mutant phenotype while clearing the mutant background. The maize material will be planted in the seedling tray on May 15, 2020, and transplanted to the experimental site of Gansu Agricultural University 10 days later (row spacing 60 cm, plant spacing 25 cm), and harvested in the middle of October 2020. When the *rpn10* grows to the V3 stage, take the mutant leaves and extract DNA according to the operating instructions of the Tiangen DNA kit. The *rpn10* mutation site used Primer Premier 5.0 software to design primers (F: CTCATATGCGGTGCCAAGAC, R: CGTGCATACAGGCGAGAATC). After PCR amplification, the product was sent to Shanghai Bioengineering Company for sequencing. The sequencing results were compared with DNAMAN6.0 software. After the end of maize powder, the phenotypic data were measured, and the seed data were counted after harvest.

### 4.2. Cytological Analysis

During the maize powder stage, paraffin sections were prepared using the lower lobe ear site of B73 and *rpn10* plants. The experimental procedure is described by Chen et al. [31]. The images were observed using a LEICA DM500 optical microscope and the ToupView camera system was used to take pictures. Measurement of leaf ear parenchyma cell size and thickness using Image-J software.

### 4.3. Data Analysis

All analyses were performed in SPSS (SPSS version 24.0, IBM SPSS statistics), statistical analysis was ANOVA (ANOVA) and post-hoc analysis was Duncan test, all plots with Origin Pro 2021 and tables with Microsoft Excel. The gene and amino acid sequences were obtained using the NCBI (https://www.ncbi.nlm.nih.gov (accessed on 22 April 2022)) database, ClustalW (https://www.ebi.ac.uk/Tools/msa/clustalw2 (accessed on 10 September 2022)) performed multiple alignments of amino acid sequences, phylogenetic trees were constructed using MEGA7.0 software.

### 4.4. ZmRPN10 Expression Pattern Analysis

RNA was extracted from roots, stems, first leaf, second leaf, and pulvinus, grown to 7d, 14d, and 21d, respectively. RNA extraction and reverse transcription were provided with the SteadyPure Plant RNA extraction kit and the Evo M-MLV reverse transcript reagent according to Ivory Biotech. Real-time fluorescence measurement was performed using the SYBR^®^ Green Pro Taq HS premixed qPCR kit, Eco (Illumina) quantitative PCR instrument amplification, the results of 2^−ΔΔCt^ calculation and analysis.

### 4.5. Exogenous BR Concentration Screening

B73 and *rpn10* seeds were soaked in distilled water for 12 h and planted in pots with supplemented vermiculite, and five seeds were seeded in each basin and repeated three times. When the seedlings were grown to V3 Stage, the roots were rinsed, stored in distilled water for 12 h, and processed in different concentrations of exogenous BR (0, 0.04, 0.2,1 and 5 μmol/L) for 12 h. Next, the second leaf angle was counted, and the optimal concentration was selected according to the change in leaf angle. Root length and seedling length were measured at different concentrations when seedlings grew to the V3 Stage.

### 4.6. Transcriptome Sequencing

CK in the control group: distilled water treatment; BR in the comparison group: 0.2 μmol/L BR. Samples were taken from the upper and lower 1 cm of the pulvinus of V3 stage of B73 and *rpn10* under CK and BR condition treatments, respectively (three replicates were set up in each group, a total of 12), quickly put into a freezing tube and put it into liquid nitrogen, and store it in a refrigerator at −70 °C. The RNA from the total samples was isolated and purified according to the protocol provided by the TRIzol (Invitrogen, CA, USA) reagent manufacturer. The amount and purity of the extracted total RNA were then quality-controlled using NanoDrop ND-1000 (NanoDrop, Wilmington, DE, USA). The total RNA of each treated sample was taken for the construction of the RNA-seq library, which was carried out by Lianchuan Bio (Hangzhou, China). The mRNA with PolyA was specifically captured by two rounds of purification using oligo (dT) magnetic beads (25–61005, Thermo Fisher Scientific, Waltham, MA, USA). The captured mRNA was fragmented under high temperature conditions and treated at 94 °C for 5–7 min. The fragmented RNA was used to synthesize cDNA. Next, Escherichia coli DNA polymerase I and RNase H were used for double-strand synthesis. These complex double strands of DNA and RNA were transformed into a DNA double-strand, a dUTP solution was incorporated into the double strand, and the ends of the double-stranded DNA were complemented to the flat ends. An A-base was then added to each end to enable it to ligate with a connector with a T-base at the end, and the fragment size was then screened and purified using magnetic beads. The second strand was digested with UDG enzyme, and then the library with a fragment size of 300 ± 50 bp was formed by PCR pre-denaturation held at 95 °C for 3 min, denaturation at 98 °C for a total of 8 cycles of 15 s each, annealing to 60 °C held for 15 s, extension at 72 °C for 30 s, and final extension held at 72 °C for 5 min. Finally, it was double-end sequenced using Illumina NovaseqTM 6000 (LC Bio Technology CO., Ltd. Hangzhou, China) in PE150 sequencing mode according to standard practice. Cutadapt software (version:cutadapt-1.9) was used to remove the reads that contained adaptor contamination. In addition, after removing the low-quality bases and undetermined bases, we used HISAT2 software (version: hisat2–2.0.4) to map reads to the genome. The mapped reads of each sample were assembled using StringTie (version:stringtie-1.3.4d) with default parameters. Next, all transcriptomes from all samples were merged to reconstruct a comprehensive transcriptome using gffcompare software (version: gffcompare-0.9.8). After the final transcriptome was generated, StringTie and ballgown were used to estimate the expression levels of all transcripts and perform expression levels for mRNAs by calculating FPKM. The differentially expressed mRNAs were selected with fold change > 2 or fold change < 0.5 and *p*-value < 0.05 by R package edgeR (https://bioconductor.org/packages/release/bioc/html/edgeR.html (accessed on 20 May 2022)), and then analysis GO enrichment and KEGG enrichment to the differentially expressed mRNAs. The original sequencing reads have been submitted to the SRA at NCBI (Accession number: PRJNA895593).

### 4.7. Co-Expression Network Analysis for Module Construction

Gene co-expression network construction was using R-package WGCNA for expression data from 12 gene samples processed by exogenous BR after B73 and *rpn10* [52]. The threshold is set as follows: gene expression mean FPKM (default: 1). Controls the similarity threshold of Module fusion (default: 0.5, the higher the Fold, the lower the similarity required for 2 modules fusion), Module member folder is determined based on kME > 0.7 (kME is the value to assess effective connectivity between key genes), the minimum number of genes within Module (default: 30). Upper limit on the number of genes displayed in the Cytoscape interaction network (default: 150). The gene expression adjacency matrix was constructed and used to analyze the network topology. In addition, modular correlation analysis was performed on module feature values and phenotypic trait data, and the correlation coefficient between phenotypic trait data and gene module trait values was calculated using Pearson correlation to map their correlation heatmaps. The OmicShare tool 2 was used to map the network visualization of the genes within the module.

## 5. Conclusions

Leaf angle is one of the most important indicators of maize plant type structure. It is of great significance for maize breeding and germplasm resources innovation to explore the genes regulating leaf angle, study the functions of genes related to leaf angle from the aspect of hormone signal transmission, and study the molecular mechanism of gene-regulating leaf angle. The phenotype and genotype of B73 and *rpn10* were identified from agronomic traits, gene identification and bioinformatics analysis. Different concentrations of exogenous BR were used to screen B73 and *rpn10* in the V3 Stage, and the optimal concentration was obtained from the hormone response level. The function of *ZmRPN10* was analyzed by transcriptome sequencing results. In the KEGG enrichment pathway, we found seven candidate genes related to leaf angle from differentially expressed genes. According to the analysis results, *ZmRPN10* mainly regulates the leaf angle of maize by regulating five kinds of hormones: CTK, ABA, BR, IAA and JA. Results showed that *ZmRPN10* might play an important role in leaf angle development. This study provides an important finding allowing further elucidation of the molecular mechanism of regulation of leaf angle in maize.

## Figures and Tables

**Figure 1 ijms-24-00189-f001:**
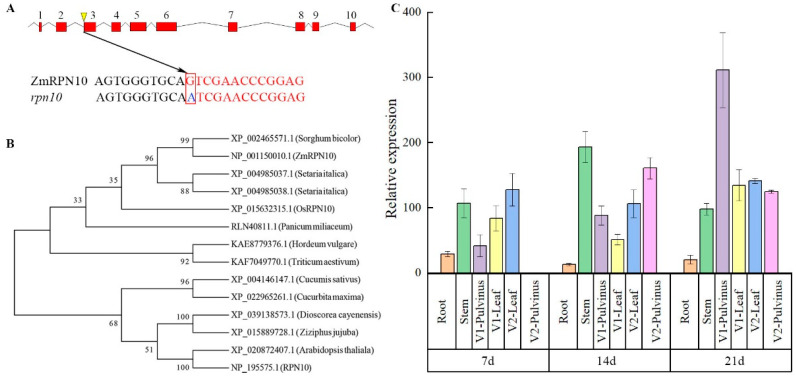
Analysis of *ZmRPN10* in Maize. (**A**) Alignment of *ZmRPN10* cDNA sequences in B73 and *rpn10*; (**B**) Phylogenetic tree of ZmRPN10; (**C**) Transcription expression of *ZmRPN10* in different tissues of maize (V1: the first leaf in the seedling stage; V2: the second leaf in the seedling stage).

**Figure 2 ijms-24-00189-f002:**
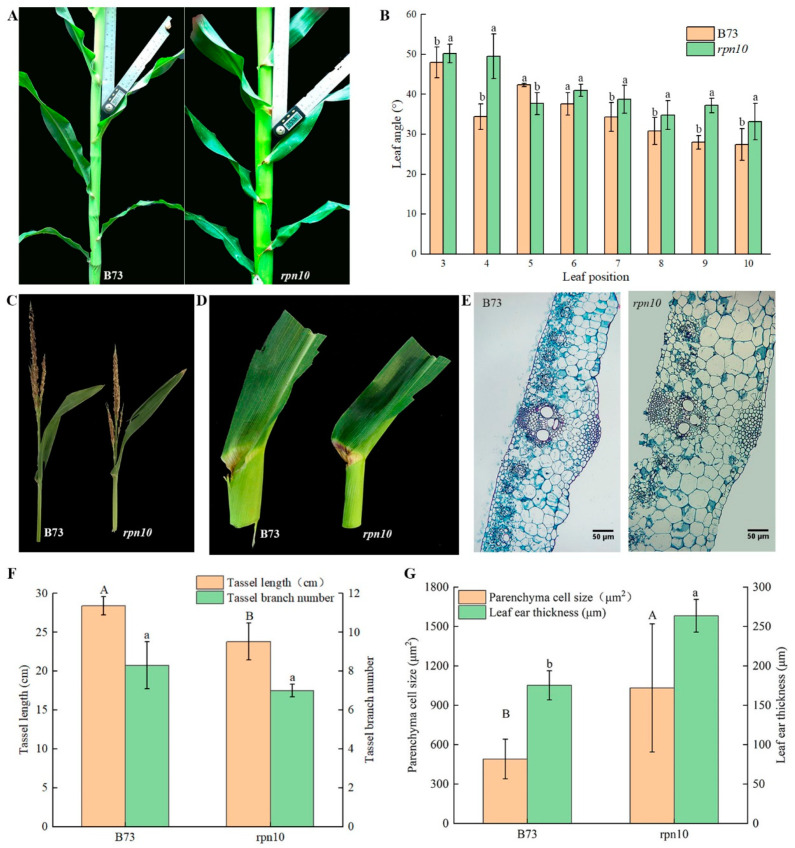
Field phenotype analysis of B73 and *rpn10* (**A**) Field morphology of B73 and *rpn10* at the powder-spreading stage; (**B**) Leaf angle between the third to the tenth leaf of the B73 and *rpn10* at the powder-spreading stage; (**C**) Field performance of B73 and *rpn10* tassels; (**D**) B73 and *rpn10* leaf ear morphology at the loosening stage; (**E**) Cross-section of leaf ear cytology at the pollen stage B73 and *rpn10*; (**F**) Tassel length and branch number of B73 and *rpn10* tassels; (**G**) Parenchyma cell size and thickness of B73 and *rpn10* leaf ear, Bars = 50 μm. Note: Upper-case and lower-case letters indicate significant differences between the yellow and green columns at 0.05 level, respectively.

**Figure 3 ijms-24-00189-f003:**
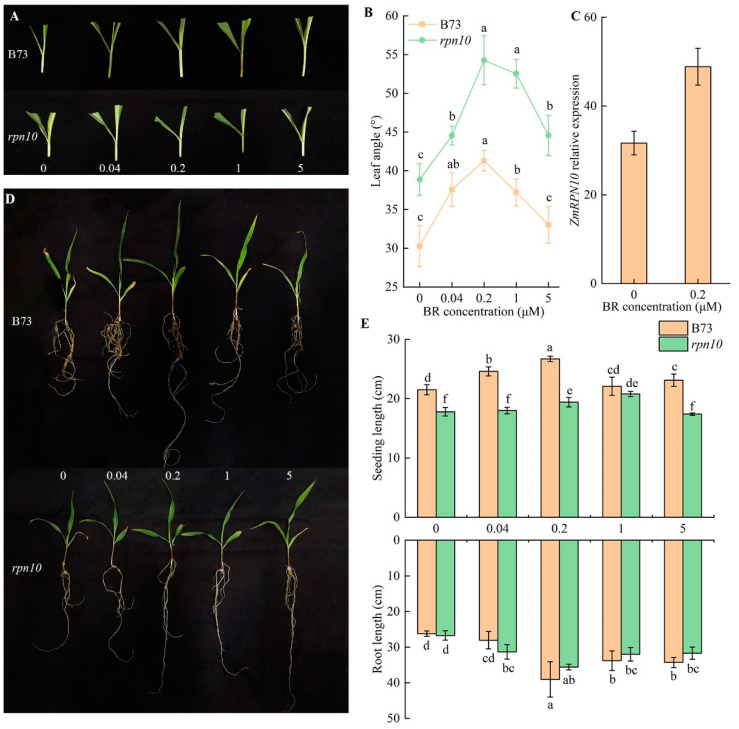
The effects of exogenous BR treatment on B73 and *rpn10* seedlings (**A**) Effects of exogenous BR treatment on leaf angle between B73 and *rpn10* seedlings; (**B**) The size of leaf angle between B73 and *rpn10* seedlings under exogenous BR treatment. Different lowercase letters on the same line indicate significant differences between treatments, *p* < 0.05; (**C**) Relative expression levels of *ZmRPN10* under 0.2 μmol/L exogenous BR treatment; (**D**) Seedling morphology of B73 and *rpn10* under different exogenous BR treatments; (**E**) Morphological changes of B73 and *rpn10* at seedling stage under different exogenous BR treatments (0 = 0 μmol/L BR, 0.04 = 0.04 μmol/L BR, 0.2 = 0.2 μmol/L BR, 1 = 1 μmol/L BR and 5 = 5 μmol/L BR). Different lowercase letters indicate significant difference between different treatments, *p* < 0.05.

**Figure 4 ijms-24-00189-f004:**
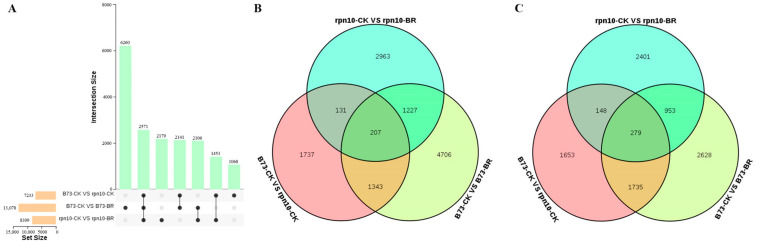
The number of differential genes under different materials and BR (**A**) Upset plot analysis of all DEGs under different materials and BR treatments; (**B**) Up-regulated DEGs identified under different materials and BR treatments; (**C**) Down-regulated DEGs identified under different materials and BR treatments. Columns represent DEG under single or multiple processes, the x-axis represents combinations of different groups (black dots represent single-treatment comparisons, black lines connected by dots represent multi-treatment comparisons), and the y-axis represents the number of genes corresponding to combinations.

**Figure 5 ijms-24-00189-f005:**
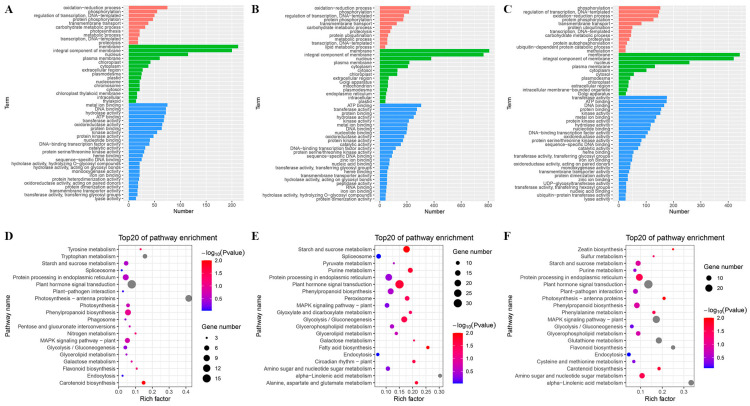
GO function and enrichment pathway analysis of differential genes in different material comparison groups. (**A**) GO analysis of differential genes in the B73-CK vs. *rpn10*-CK comparison group; (**B**) GO analysis of differential genes in the B73-CK vs. B73-BR comparison group; (**C**) GO analysis of differential genes in the *rpn10*-CK vs. *rpn10*-BR comparison group (Red indicates biological process; green indicates cellular component; blue indicates molecular function); (**D**) significant pathway enrichment analysis of B73-CK vs. *rpn10*-CK comparison group; (**E**) significant pathway enrichment analysis in the B73-CK vs. B73-BR comparison group; (**F**) significant pathway enrichment analysis in the *rpn10*-CK vs. *rpn10*-BR comparison group.

**Figure 6 ijms-24-00189-f006:**
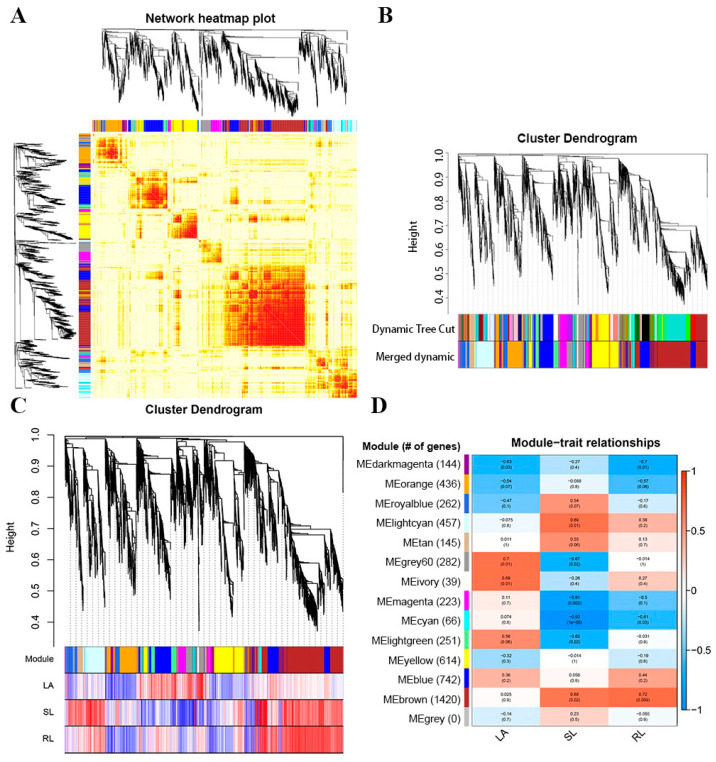
Expression level cluster analysis and phenotypic association. (**A**) Heatmap of the gene co-expression network. (**B**) Dendrogram and module color mixing plot after merging module. (**C**) The tom dendrogram and the heatmap of the correlation between genes and samples. (**D**) Heatmap of the correlation between module and sample type (grey modules are unassigned genes module). Note: LA, Leaf angle; SL, seeding length; RL, root length.

**Figure 7 ijms-24-00189-f007:**
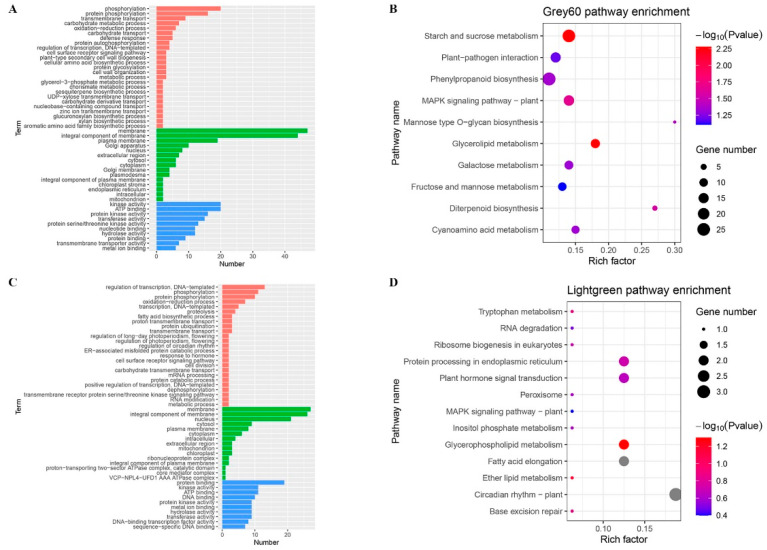
The functional analysis of genes in the phenotypic significant enrichment module. (**A**) GO analysis of differential genes in grey60 module (Red indicates biological process; green indicates cellular component; blue indicates molecular function); (**B**) significant pathway enrichment analysis in grey60 module; (**C**) GO analysis of differential genes in the light green module; (**D**) significant pathway enrichment analysis in the light green module.

**Figure 8 ijms-24-00189-f008:**
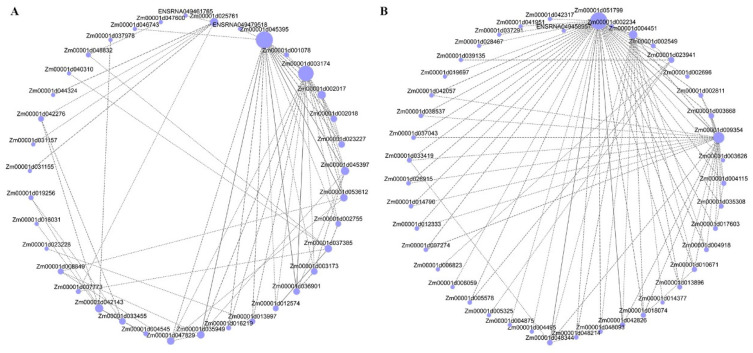
Analysis of hub genes network interaction in phenotypic significant enrichment module. (**A**) Network interaction analysis of hub genes in grey60 module. (**B**) Network interaction analysis of hub genes in lightgreen module. The size gradient of the points represents the level of the soft threshold of connectivity.

**Figure 9 ijms-24-00189-f009:**
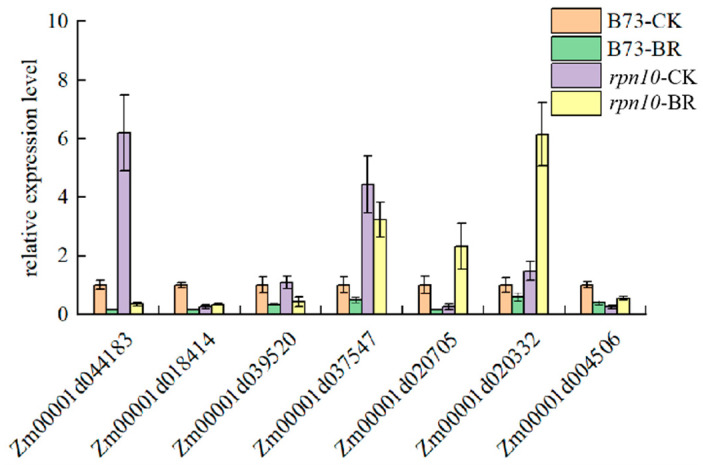
The validation of relative expression of selected genes by qRT-PCR.

**Figure 10 ijms-24-00189-f010:**
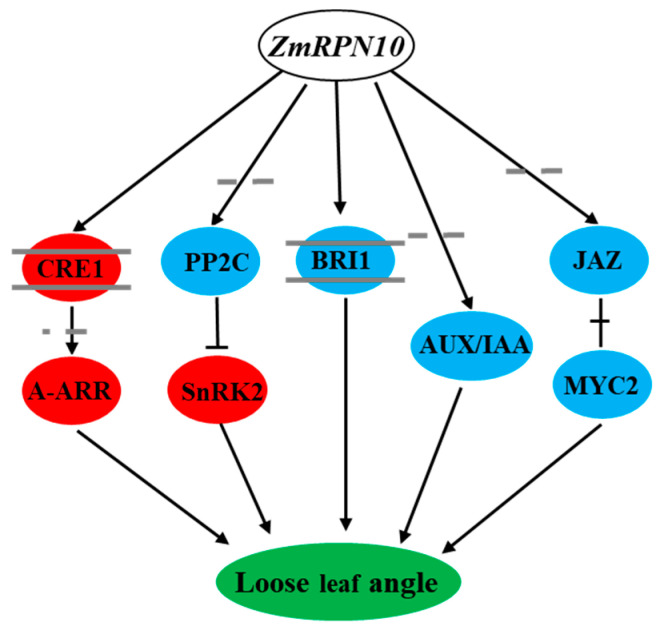
The schematic model of *ZmRPN10* regulating maize leaf angle. The red circle indicates that the expression of related genes in *rpn10* is higher than B73, and the blue circle indicates that the expression of related genes in *rpn10* is lower than B73.

**Table 1 ijms-24-00189-t001:** The investigation on agronomic characters of B73 and *rpn10*.

Traits	B73	*rpn10*
Plant height/cm	230.7 ± 4.3 a	207.7 ± 3.9 b
Ear height	106.0 ± 1.6 a	91.3 ± 5.0 b
Stem diameter/mm	20.0 ± 0.5 a	23.8 ± 3.2 a
Leaf length/cm	80.9 ± 1.3 a	85.4 ± 1.6 a
Leaf width/cm	8.1 ± 0.2 b	9.6 ± 0.2 a
Leaf orientation value	82.0 ± 2.9 a	73.2 ± 2.3 a
Leaf size/cm^2^	488.3 ± 4.9 b	612.2 ± 11.6 a
Ear length/cm	13.2 ± 0.2 a	11.6 ± 0.90 b
Ear coarse/cm	4.1 ± 0.05 a	4.4 ± 0.4 a
Long bald/cm	2.6 ± 0.11 a	1.7 ± 0.37 b
Ear weight/g	95.0 ± 5.91 a	81.0 ± 14.8 a
100-kernel weight	24.7 ± 1.54 a	20.6 ± 2.95 a
Ear rows	18.7 ± 1.15 a	16.7 ± 1.15 a
Number of rows	23.7 ± 0.57 a	24.3 ± 3.51 a
Shaft coarse /cm	2.7 ± 0.11 b	3.2 ± 0.11 a
Shaft weight/g	20.2 ± 1.08 a	13.7 ± 2.26 b

Note: The different lowercase letters “a, b, c” indicate significant differences between different materials, *p* < 0.05.

## Data Availability

Not applicable.

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
