# Peer review of "Study on ZmRPN10 Regulating Leaf Angle in Maize by RNA-Seq"

_ijms, 2022, doi:10.3390/ijms24010189_

Round 1

Reviewer 1 Report

The authors of this manuscript studied the phenotype changes between B73 and rpn10 maize, and also performed transcriptomic analysis between these two genotypes under control and BR-treated conditions. The authors identified thousands of differentially expression genes that could be potential candidate contributing to the observed leaf angle changes. However, the authors failed to deliver a strong argument for their hypothetical models. The general logical flow of manuscript is: the authors identified the candidate genes in rpn10 mutant (which was demonstrated to show leaf angle change), the authors described the functions of those candidates, and the authors proposed the candidates controlling leaf angles. However, I didn't see the connection between the gene function and leaf angles. Many genes, e.g. genes involved in hormonal signaling, could participate a broad category of biological processes, and therefore it is very hard to conclude that they have a direct control over leaf angles or specifically control the leaf angles. The same comment applies to the functional enrichment analysis, the significantly enriched GO terms are too broad to make any meaningful conclusion regarding their role in leaf angle changes. Finally, RPN10 encodes a subunit of 26 proteasome that is an essential part of a post-translational regulatory mechanism, the ubiquitin-proteasome system. But the authors didn't provide any information regarding the post-translational regulation, which makes one wonder what is the rationale of studying RPN10. More detailed comments are provided as follows:

1. Fig 2 and Table 1: Fig.2 shows a difference in morphology between B73 and rpn10. However, as no statistical support has been provided for this figure, it's hard to know whether this difference is statistically significant. Actually, based on the error bars presented in the figures, it's quite possible that such difference is NOT significant. The authors did provide statistical evidence in Table 1, but failed to mention the hypothesis testing that generated the evidence.

The same comment applies to fig.3.

2. How many genes were expressed in total in B73 and rpn10?

3. Fig.6, what are "LA", “SL“, and "RL"? Based on fig. 6D, module ivory also appeared to be highly correlated with leaf angle change (actually even more correlated than module lightgreen), why is it not mentioned in the manuscript?

4. line 299- line 308: The authors cited two articles wherein maize mutants showed phenotype changes in grain size and lignin compositions. However, as this study is focused on the leaf angle changes, I cannot see why these two studies were cited. There are tons of studies showing phenotype changes caused by mutations in maize, why specifically cite these two? What is the connection between these two and the current study?

5. line 330 - 362 and figure 9: the authors selected seven candidate genes based on DEG analysis between B73 and rpn10 under BR-treated and control conditions. However, as shown in fig.4, thousands of genes were differentially expressed, what is the rationale of selecting only seven candidates? Is there anything special about these seven candidates? How were they selected (i.e., by what standards?) 

Moreover, based on the discussion of functions in these seven candidate genes, it is very hard to tell the message that the authors trying to convey. How are they associated with leaf angle change? Are there any prior studies demonstrating that they are associated with leaf angle change?

The similar comment applies to the next section, when the authors discuss the roles of DEGs between B73 and rpn10. Again, the authors "cherry picked" a few genes out of thousands of DEGs and described their functions, but failed to make connection between the functions and leaf angle changes. 

6. RNA-seq analysis were poorly described in the method section, how many samples were used for the analysis? (i.e., How many conditions? How many biological replicates were provided for each condition)? How was the total RNA extracted? How were they purified? How were the RNA-seq libraries prepared? Which instrument was used for RNA-seq? Was it single-end or pair-end sequencing?

After obtaining the raw reads data, how were the RNA-seq datasets analyzed? What programs were used for data cleaning, read alignment, assembly and quantification? 

These are all necessary information for methods but not mentioned in the study at all.

Author Response

Response to Reviewer 1 Comments

Dear reviewer:

We appreciate these valuable comments and suggestions very much. We have made detailed corrections in the manuscript corresponding to your suggestions and advice. Thank you for your time and consideration. According to the reviewers' comments, we have seriously revised those problems. At the same time, we have also made other changes, and marked the revised and supplemented sections in red font.

Point 1: Fig 2 and Table 1: Fig.2 shows a difference in morphology between B73 and rpn10. However, as no statistical support has been provided for this figure, it's hard to know whether this difference is statistically significant. Actually, based on the error bars presented in the figures, it's quite possible that such difference is NOT significant. The authors did provide statistical evidence in Table 1, but failed to mention the hypothesis testing that generated the evidence. The same comment applies to Fig.3.

Response 1: Thanks for your important comment and suggestion. We reanalyzed the relevant data in Figures 2, Figure 3, and Table 1, using a statistical analysis of variance (ANOVA) and a post-hoc analysis with the Duncan test, complementing the experimental method in paragraph 4.3.

Point 2: How many genes were expressed in total in B73 and rpn10?

Response 2: Thanks for your important comment and suggestion. A total of 17,778 genes were expressed in B73 and rpn10.

Point 3: Fig.6, what are "LA", “SL“, and "RL"? Based on fig. 6D, module ivory also appeared to be highly correlated with leaf angle change (actually even more correlated than module lightgreen), why is it not mentioned in the manuscript?

Response 3: Thanks for your important comment and suggestion. ‘LA’ for leaf angle, ‘SL’ for seeding length, and ‘RL’ for root length, we have annotated it in Fig.6. Although observed from Figure 6D, the module tusk is highly correlated with the changes in the leaf angle, the number of genes in this module is too small, and we did not analyze this module gene during the later analysis process, so we did not mention it. We have made the L-241 supplement.

Point 4: line 299- line 308: The authors cited two articles wherein maize mutants showed phenotype changes in grain size and lignin compositions. However, as this study is focused on the leaf angle changes, I cannot see why these two studies were cited. There are tons of studies showing phenotype changes caused by mutations in maize, why specifically cite these two? What is the connection between these two and the current study?

Response 4: Thanks for your important comment and suggestion. We cited these two articles mainly to show that maize mutants can cause phenotypic changes in maize, and then we also added literature related to changes in leaf angle caused by mutations to the article, and modified the subtitle.

Point 5: line 330 - 362 and figure 9: the authors selected seven candidate genes based on DEG analysis between B73 and rpn10 under BR-treated and control conditions. However, as shown in fig.4, thousands of genes were differentially expressed, what is the rationale of selecting only seven candidates? Is there anything special about these seven candidates? How were they selected (i.e., by what standards?)

Moreover, based on the discussion of functions in these seven candidate genes, it is very hard to tell the message that the authors trying to convey. How are they associated with leaf angle change? Are there any prior studies demonstrating that they are associated with leaf angle change?

The similar comment applies to the next section, when the authors discuss the roles of DEGs between B73 and rpn10. Again, the authors "cherry picked" a few genes out of thousands of DEGs and described their functions, but failed to make connection between the functions and leaf angle changes.

Response 5: Thanks for your important comment and suggestion. In Figure 9, we randomly selected seven genes for fluorescence quantification, and the purpose was to verify the reliability of the transcriptome sequencing results, so these seven genes are not related to the following seven candidate genes. In paragraph 3.3 (line 330-362) we selected seven candidate genes for based on the results of pathway enrichment analysis and cluster transcriptome sequencing, under | log2FC |> 1 and p <0.001 under BR treatment and control conditions. In addition, we reviewed and summarized the functions of these seven candidate genes.

Point 6: RNA-seq analysis were poorly described in the method section, how many samples were used for the analysis? (i.e., How many conditions? How many biological replicates were provided for each condition)? How was the total RNA extracted? How were they purified? How were the RNA-seq libraries prepared? Which instrument was used for RNA-seq? Was it single-end or pair-end sequencing?

After obtaining the raw reads data, how were the RNA-seq datasets analyzed? What programs were used for data cleaning, read alignment, assembly and quantification?

Response 6: Thanks for your important comment and suggestion. We have redescribed the RNA-seq analysis in paragraph 4.6.

Thank you again for your detailed and significant suggestions. Based on your comments, we have revised the corresponding content and grammatical in the manuscript and hope that the correction will meet with your approval.

Best wishes!

Reviewer 2 Report

The figure 1 in supplementary document can be added in the manuscript. 

Author Response

Response to Reviewer 2 Comments

Dear reviewer:

We appreciate these valuable comments and suggestions very much. We have made detailed corrections in the manuscript corresponding to your suggestions and advice. Thank you for your time and consideration. According to the reviewers' comments, we have seriously revised those problems. At the same time, we have also made other changes, and marked the revised and supplemented sections in red font.

Point 1:  The figure 1 in supplementary document can be added in the manuscript.

Response 1: Thanks for your important comment and suggestion. We think your suggestion is very good. We added Supplementary Fig.1 to the Fig.3C position of the manuscript.

Thank you again for your detailed and significant suggestions. Based on your comments, we have revised the corresponding content in the manuscript and hope that the correction will meet with your approval.

Best wishes!

Reviewer 3 Report

The presented manuscript is devoted to the study of the angle of inclination of the maize leaf. The amount of research carried out is large, but it does not reveal the mechanisms of regulation of the angle of inclination of the leaf, but only reveals some possible participants in the regulation process, and how the regulation process itself proceeds remains unknown. In the future, these results may be the basis for studying the mechanism of tilt angle regulation. In this regard, I recommend changing the title of the manuscript.

The results of the experiment are interesting, there is no doubt about the data obtained. However, their discussion causes some bewilderment. In paragraph 3.1. the description of the literature data does not correspond to the title of the subtitle. In paragraph 3.2. the discussion of the results presented is not relevant to the description of the mechanism of action. In paragraph 3.3. no discussion of the effect of BR action.

A small note on the design of the manuscript. In Fig. 6, increase the captions at the top graphs

Author Response

Response to Reviewer 3 Comments

Dear reviewer:

We appreciate these valuable comments and suggestions very much. We have made detailed corrections in the manuscript corresponding to your suggestions and advice. Thank you for your time and consideration. According to the reviewers' comments, we have seriously revised those problems. At the same time, we have also made other changes, and marked the revised and supplemented sections in red font.

Point 1: In this regard, I recommend changing the title of the manuscript.

Response 1: Thanks for your important comment and suggestion. We have changed the title to ‘Study on ZmRPN10 Regulating Leaf Angle in Maize by RNA-Seq’.

Point 2: In paragraph 3.1. the description of the literature data does not correspond to the title of the subtitle.

Response 2: Thanks for your important comment and suggestion. We cited these two articles mainly to show that maize mutants can cause phenotypic changes in maize. We have modified the subtitle in paragraph 3.1 to match it with the description of the literature data.

Point 3: In paragraph 3.2. the discussion of the results presented is not relevant to the description of the mechanism of action.

Response 3: Thanks for your important comment and suggestion. We modified the title of paragraph 3.2 to correspond with the discussion of the results stated.

Point 4: In paragraph 3.3. no discussion of the effect of BR action.

Response 4: Thanks for your important comment and suggestion. The title of paragraph 3.3 has been modified to reflect the discussion on the reported results, and the content of the discussion has also been appropriately revised.

Point 5: A small note on the design of the manuscript. In Fig. 6, increase the captions at the top graphs

Response 5: Thanks for your important comment and suggestion. We have increased the captions at the top graphs in Fig. 6.

Thank you again for your detailed and significant suggestions. Based on your comments, we have revised the corresponding content in the manuscript and hope that the correction will meet with your approval.

Best wishes!

Reviewer 4 Report

The work of the authors is very interesting, and definitely deserves attention.
However, several questions arise. Firstly, what exactly was the statistical method of analysis used by the authors, what was the post-hoc?
Second, in Figure 2F, I see a difference in morphology. Why the authors did not carry out morphometry of microscopic images. This will allow us to speak about the influence of the effects observed by the authors on cell sizes, namely, that growth occurs by stretching, and not by increasing the number of cells, if this parameter is reliable. I strongly recommend that the authors do this calculation and include this data in the article, which will allow hims to talk about cellular effects on the size of the whole plant.

Author Response

Response to Reviewer 4 Comments

Dear reviewer:

We appreciate these valuable comments and suggestions very much. We have made detailed corrections in the manuscript corresponding to your suggestions and advice. Thank you for your time and consideration. According to the reviewers' comments, we have seriously revised those problems. At the same time, we have also made other changes, and marked the revised and supplemented sections in red font.

Point 1: What exactly was the statistical method of analysis used by the authors, what was the post-hoc?

Response 1: Thanks for your important comment and suggestion. The statistical analysis method we used was Analysis of Variance (ANOVA), and the post-hoc analysis method was Duncan test. In paragraph 4.3 added.

Point 2: In Figure 2F, I see a difference in morphology. Why the authors did not carry out morphometry of microscopic images. This will allow us to speak about the influence of the effects observed by the authors on cell sizes, namely, that growth occurs by stretching, and not by increasing the number of cells, if this parameter is reliable. I strongly recommend that the authors do this calculation and include this data in the article, which will allow hims to talk about cellular effects on the size of the whole plant.

Response 2: Thanks for your important comment and suggestion. Since we are using a normal microscope when looking at cell morphology, we cannot measure the size of the cells now (Due to the impact of COVID-19, we were not able to test again to measure cell size recently).

Thank you again for your detailed and significant suggestions. Based on your comments, we have revised the corresponding content in the manuscript and hope that the correction will meet with your approval.

Best wishes!

Round 2

Reviewer 1 Report

In general most of my comments are properly addressed. However, there are a few remaining points that can be further improved. Here are my responses towards these remaining points.

"Response 3: Thanks for your important comment and suggestion. ‘LA’ for leaf angle, ‘SL’ for seeding length, and ‘RL’ for root length, we have annotated it in Fig.6. Although observed from Figure 6D, the module tusk is highly correlated with the changes in the leaf angle, the number of genes in this module is too small, and we did not analyze this module gene during the later analysis process, so we did not mention it. We have made the L-241 supplement."

If this is the case for module ivory, please consider marking the number of genes in each module in the graph and clarify the rationale in the text.

"Response 5: Thanks for your important comment and suggestion. In Figure 9, we randomly selected seven genes for fluorescence quantification, and the purpose was to verify the reliability of the transcriptome sequencing results, so these seven genes are not related to the following seven candidate genes. In paragraph 3.3 (line 330-362) we selected seven candidate genes for based on the results of pathway enrichment analysis and cluster transcriptome sequencing, under | log2FC |> 1 and p <0.001 under BR treatment and control conditions. In addition, we reviewed and summarized the functions of these seven candidate genes."

How many DEGs satisfy the criteria of |log2FC|>1 and p <0.001, are there only seven genes? In line 345-346, the authors stated that these seven genes are related to leaf angle. However, the functions of these genes, as described by the authors, have a quite general role in plant development and I cannot see why they are specifically linked to leaf angle. Moreover, in line 369-371, the authors stated "How the 369 new genes discovered in the study respond to exogenous hormones to regulate the devel- 370 opmental mechanism of leaf angles remains to be further studied", which appears to contradict their statement in line 345-346. Similarly, the authors proposed a hypothetical pathway model for ZmRPN10 regulating leaf angles, however the genes included in the model have too general a function in regulating plant growth and I couldn't see how this model is specific for leaf angle.

In addition, in line 323-337, the authors cited a few maize genes (e.g., ZmBEH1, YABBY, ZmLG2) that have been demonstrated to control leaf angles, what about their expression pattern (in BR-treated and control conditions) in this study?

Author Response

Response to Reviewer 1 Comments

Dear reviewer:

We appreciate these valuable comments and suggestions very much. We have made detailed corrections in the manuscript corresponding to your suggestions and advice. Thank you for your time and consideration. According to the reviewers' comments, we have seriously revised those problems. At the same time, we have also made other changes, and marked the revised and supplemented sections in red font.

Point 1: If this is the case for module ivory, please consider marking the number of genes in each module in the graph and clarify the rationale in the text.

Response 1: Thanks for your important comment and suggestion. We determined the number of members per module based on the kME values, and we have added the number of genes per module in Figure 6D and illustrated the module gene selection criteria in paragraph 4.7.

Point 2: How many DEGs satisfy the criteria of |log2FC|>1 and p <0.001, are there only seven genes? In line 345-346, the authors stated that these seven genes are related to leaf angle. However, the functions of these genes, as described by the authors, have a quite general role in plant development and I cannot see why they are specifically linked to leaf angle. Moreover, in line 369-371, the authors stated "How the 369 new genes discovered in the study respond to exogenous hormones to regulate the devel- 370 opmental mechanism of leaf angles remains to be further studied", which appears to contradict their statement in line 345-346. Similarly, the authors proposed a hypothetical pathway model for ZmRPN10 regulating leaf angles, however the genes included in the model have too general a function in regulating plant growth and I couldn't see how this model is specific for leaf angle.

Response 2: Thanks for your important comment and suggestion. We screened the DEGs using the criteria of | log2FC |> 2 and p <0.001, yielding a total of 18 genes and discussed in paragraph 3.3. And the pathway model of ZmRPN10 regulating leaf angle was reillustrated in L411-413.

Point 3: In addition, in line 323-337, the authors cited a few maize genes (e.g., ZmBEH1, YABBY, ZmLG2) that have been demonstrated to control leaf angles, what about their expression pattern (in BR-treated and control conditions) in this study?

Response 3: Thanks for your important comment and suggestion. The expression patterns of the cited maize genes (ZmBEH1, yabbby, ZmLG 2) (in BR-treated and control conditions) were described at L334-338.

Thank you again for your detailed and significant suggestions. Based on your comments, we have revised the corresponding content and grammatical in the manuscript and hope that the correction will meet with your approval.

Best wishes! 

Reviewer 4 Report

The authors have pictures, they showed them in the article. It does not matter on which microscope the work was done, if there are photographs, a morphometric analysis can be done. To do this, you don’t even need to have software for a microscope, but you can do it on any computer in a free program, such as Image-J. Therefore, the answer of the authors about the unavailability of the equipment does not look convincing, and I will insist on performing a morphometric analysis, since this will improve the quality of the article and make the observations more complete and the conclusions justified.

Author Response

Response to Reviewer 4 Comments

Dear reviewer:

We appreciate these valuable comments and suggestions very much. We have made detailed corrections in the manuscript corresponding to your suggestions and advice. Thank you for your time and consideration. According to the reviewers' comments, we have seriously revised those problems. At the same time, we have also made other changes, and marked the revised and supplemented sections in red font.

Point 1: The authors have pictures, they showed them in the article. It does not matter on which microscope the work was done, if there are photographs, a morphometric analysis can be done. To do this, you don’t even need to have software for a microscope, but you can do it on any computer in a free program, such as Image-J. Therefore, the answer of the authors about the unavailability of the equipment does not look convincing, and I will insist on performing a morphometric analysis, since this will improve the quality of the article and make the observations more complete and the conclusions justified.

Response 1: Thanks for your important comment and suggestion. We have measured the leaf ear parenchyma cell size and thickness using the Image-J software, the contents of paragraph 2.2 have been also appropriately modified, with the measurement data added to Figure 2G, and the measurement method added to paragraph 4.2.

Thank you again for your detailed and significant suggestions. Based on your comments, we have revised the corresponding content in the manuscript and hope that the correction will meet with your approval.

Best wishes!

Round 3

Reviewer 4 Report

Dear authors, thank you for making changes to the article, you really see that the data changes make the article more complete.